# The Effects of Temperature on the Development, Morphology, and Fecundity of *Aenasius bambawalei* (=*Aenasius arizonensis*)

**DOI:** 10.3390/insects12090833

**Published:** 2021-09-16

**Authors:** Juan Zhang, Yayuan Tang, Jun Huang

**Affiliations:** 1Zhejiang Institute of Landscape Plants and Flowers/Zhejiang Xiaoshan Institute of Cotton & Bast Fiber Crops, Hangzhou 311251, China; juanjuan0031@aliyun.com (J.Z.); yayuan1127@163.com (Y.T.); 2State Key Laboratory for Managing Biotic and Chemical Threats to the Quality and Safety of Agro-Products, Institute of Plant Protection and Microbiology, Zhejiang Academy of Agricultural Sciences, Hangzhou 310021, China

**Keywords:** development, hind tibia of female adults, fecundity, *Aenasius bambawalei*, accompanying natural enemy

## Abstract

**Simple Summary:**

During biological invasions, insect pest outbreaks often occur because they escape the control of natural enemies from their place of origin. However, some natural enemies can migrate with pests to effectively inhibit their damage. Whether temperature changes can shorten or enhance the reproductive developmental period of accompanying natural enemies is an important determinant of whether they can effectively migrate with pests. *Aenasius bambawalei* is a predominant accompanying parasitoid of the important invasive pest *Phenacoccus solenopsis*. To understand the effect of temperature changes on the development of the reproductive systems of this parasitoid, we compared differences in its development, morphology, and fecundity under different temperatures. The results showed that high temperature could significantly shorten the pupal developmental duration, increase the length of the hind tibia, and accelerate gonad development. Our results indicated that moderately high temperature was conducive to the reproduction and development of the parasitoid and may be related to the more females offspring. This is the first report of the impact of high temperature on the pupal development, morphology, and fecundity of *A. bambawalei*.

**Abstract:**

The effects of high temperature on the developmental, morphological, and fecundity characteristics of insects, including biological invaders and their accompanying natural enemies, are clear. *Phenacoccus solenopsis* (Homoptera: Pseudococcidae) is an aggressive invasive insect pest worldwide. *Aenasius bambawalei* (=*Aenasius arizonensis* Girault) (Hymenoptera: Encyrtidae) is a predominant accompanying parasitoid of this mealybug. Our previous studies showed that temperature change induced an increase in the female offspring ratio of *A. bambawalei*. However, whether this increase is the result of a shortened or enhanced development period of the reproductive systems of *A. bambawalei* remains unknown. Here, we compared the pupal development, hind tibia of female adults, and fecundity of *A. bambawalei* under different temperatures to clarify the development and morphological changes induced by high temperature and to better understand its potential as an accompanying natural enemy. Our results showed that, at a high temperature (36 °C), the pupal developmental duration of *A. bambawalei* was only 0.80 times that of the control, and the length of the hind tibia was 1.16 times that of the control. Moreover, high temperature accelerated the developmental rate of gonads and increased the numbers of eggs and sperm. These results indicated that experimental warming shortened the pupal developmental duration, altered the hind tibia length of female adults, and facilitated the fecundity of *A. bambawalei*. These findings will help to understand the adaptation mechanisms of accompanying natural enemies. Furthermore, these findings will help to make use of this behavior to effectively control invasive pests.

## 1. Introduction

High temperature may directly or indirectly influence the spread of alien species [1]. Due to differences in their capacity to respond to temperature, the migration and colonization ability of exotic natural enemies is usually not as good as that of invasive insects. As a result, invasive insects often escape the control of natural enemies in newly invaded areas [2,3]. However, natural enemies may evolve to synchronize migration with their host [4,5,6]. In this process, some adaptive developmental, morphological, and fecundity changes may occur in the accompanying natural enemies [5]. Clarifying these changes of accompanying natural enemies to high temperature is therefore of important ecological and behavioral significance [7]. For exotic natural enemies, most studies have focused on the risk assessment of nonnative control agents [8,9] but not on the developmental, morphological, or fecundity changes resulting from elevated temperatures [10].

For alien species, changes in development, morphology and fecundity at high temperatures are exhibited in various aspects [11,12]. The cotton mealybug *Phenacoccus solenopsis* Tinsley (Hemiptera: Pseudococcidae) has been recognized as an aggressive invasive insect pest worldwide [13]. In some countries, this mealybug causes severe economic damage to cotton and to a wide range of vegetables, horticultural plants and other field crops [14]. At present, many studies have reported the effects of temperature on the survival rate [15], development [16], and reproduction [17] of *P. solenopsis*. These studies showed that *P. solenopsis* had strong temperature adaptability [18]. *Aenasius bambawalei* Hayat (Hymenoptera: Encyrtidae) plays a significant role in controlling the mealybug population [19,20,21] and can migrate with *P. solenopsis* to many areas [22]. However, in some colder areas where *P. solenopsis* had recently invaded, the natural parasitism rate of *A. bambawalei* was significantly lower [23]. Our previous studies indicated that high temperature (36 °C) in the laboratory resulted in an increase in the female offspring ratio of *A. bambawalei*, and this result was even greater than that in its place of origin [24]. Thus, we hypothesized that high temperature may cause changes in the development, morphology, and fecundity of *A. bambawalei*.

To verify this hypothesis, we compared the pupal development, hind tibia of female adults, and fecundity of *A. bambawalei* at 27 °C and 36 °C. The results of this study will contribute to clarifying the adaptive mechanism to high temperature of this natural enemy and also further reveal the mechanism by which temperature changes affect the interaction between invasive insects and their exotic natural enemies.

## 2. Materials and Methods

### 2.1. Insects and Breeding Conditions

*P. solenopsis* was field-collected from *Hibiscus mutabilis* grown in Hangzhou City, Zhejiang Province, China (30°18′75″ N; 120°28′60″ E) in July 2012. In the laboratory, *P. solenopsis* was reared on cotton plants in a growth chamber (27–30 °C, 50–70% RH, and an L16:D8 photoperiod) according to the procedures of our previous experiments [25]. The colony was maintained until it reached more than 30 generations in the present study. *A. bambawalei* were reared on their host *P. solenopsis* in the lab under the same controlled conditions as *P. solenopsis*. Third-instar nymphs and female adults of *P. solenopsis* were used in the experiments because of their relatively high survival rate and suitable duration of development. To maintain genetic stability and fitness of the population, *A. bambawalei* was collected from the wild and added to lab colonies irregularly.

### 2.2. Temperature Treatments

To evaluate the effect of temperature change on the development and reproductive traits of *A. bambawalei*, it was necessary to select a suitable temperature and instar that showed a significant change [10,11]. According to previous studies by the authors [24] and other researchers [26], a 36 °C exposure for 24 h was imposed on the early pupal stage of *A. bambawalei* in this study. All wasp pupae (approximately 7 days after the mealybug was parasitized) were divided into 2 groups: (1) pupae were collected, placed separately in 1.5-mL centrifuge tubes, and then maintained in the growth chamber at 27 °C until emergence (control individuals;, or (2) pupae were placed separately in a 1.5-mL centrifuge tube and heat-treated at 36 °C for 24 h in a climate-controlled incubator and then transferred to the climate room at 27 °C until emergence (individuals on high temperature). Then, the emergence of *A. bambawalei* from these two groups was observed daily. The emergence time, number of parasitized *P. solenopsis*, and number of emerged *A. bambawalei* were recorded. Finally, as an important index to evaluate the size and fecundity of parasitoids [27], the hind tibia length of newly emergent *A. bambawalei* was measured under a Nikon SMZ1500 zoom stereomicroscope (Nikon, Tokyo, Japan). Each emergence time, emergence rate, and hind tibia length test included 60, 60, and 30 replicates, respectively.

### 2.3. Effects of Temperature on Ovarian and Sperm Development

Newly emergent (within 12 h) *A. bambawalei* were fed 10% honeydew solution, which was renewed every day until dissection. From 1 d after emergence, approximately 30 virgin females or virgin males per treatment group per day were dissected on average. The dissection was carried out according to the following steps: (1) the parasitoids were placed in the freezer (average temperature, −20 °C) for approximately 2 min to freeze; and (2) the frozen parasitoids were immersed in PBS phosphate buffer solution and dissected under an anatomical microscope (Nikon SMZ-10) to detect intact or impaired female or male gonads. Following dissection, the ovaries per female or the testes per male were observed separately using a Canon D60 digital camera coupled to the microscope. The length of ovarioles was measured, and the ovariole length per female was calculated (mm). For ovariole length (mm), based on repeat measurements of a random subset of 30 individuals, the measurement error was very small. The numbers of oocyte stalk hook analogues and mature oocytes in the ovary were counted, and the mean number per ovariole was calculated. The diameter of all mature oocytes was measured, and each of the 3 largest mature oocytes was selected as the indicator (mm). These measurements were performed under a Nikon SMZ1500 zoom stereomicroscope (Nikon, Tokyo, Japan) and used to determine the mean oocyte size per female.

As larger organs contain more sperm [28] and the sperm in the seminal vesicle are mature [29], the size of the seminal vesicle was measured under a Nikon SMZ1500 zoom stereomicroscope (Nikon, Tokyo, Japan) to evaluate testis development. In our preliminary experiments, we found that the longevity of *A. bambawalei* in incubators at 27 °C and 36 °C was up to 16 days; however, on the 12th day, the gonads began to degenerate. Thus, in this study, the data were analyzed at 12 days. Ovaries and testes in the control and high-temperature wasps were measured on the day of emergence, at 3 days old, at 5 days old, at 7 days old, and at 12 days old, and each treatment and control included 30 replicates. The criterion for male sterility was complete absence of sperm in the seminal vesicle or in the spermatheca of the mated female [29].

### 2.4. Statistical Analyses

The emergence rate is the percentage of the number of emerged *A. bambawalei* in the number of parasitized *P. solenopsis*. The effect of high temperature on pupal development, hind tibia of female adults, and reproductive parameters was determined by one-way ANOVA with appropriate post hoc tests. Student’s *t*-test was used to compare the value of means from two samples, and the chi-square test was used to compare proportions.

Ovary and test dynamics were analyzed by a generalized linear model (GLM) using R statistical software (R Development Core Team 2007, Auckland, New Zealand). Mean values of oocyte number, egg, testis, and seminal vesicle size were compared using one-way ANOVA. Post hoc comparisons for all treatments were performed using a multiple comparison test based on Student’s *t*-statistics with a Bonferroni correction. Mean values of female fecundity and egg viability were compared using a Mann-Whitney U test.

## 3. Results

### 3.1. Effects of High Temperature on Pupal Development and the Hind Tibia of Female Adults

High temperature could significantly affect the pupal developmental duration (emergence time) (*F*_(1,118)_ = 13.50, *p* < 0.05) (Figure 1A) and the hind tibia length of female adults (*F*_(1,58)_ = 22.63, *p* < 0.05) (Figure 1C) but had no effect on the emergence rate (*F*_(1,8)_ = 0.40, *p* = 0.54) (Figure 1B). Moreover, the pupal development duration (emergence time) was only 0.80 times that of the control, and the length of the hind tibia was 1.16 times that of the control. These results showed that high-temperature treatment accelerates pupal development.

### 3.2. Effects of High Temperature on Ovarian Development

Females had a pair of ovaries, and each ovary contained three ovarioles (Figure 2A–J). The ovarioles were nearly transparent, and each ovariole contained linear arrays of progressively developing follicles starting with dividing germ cells and ending with mature oocytes ready for fertilization (Figure 2A–J).

In the early stages of emergence, there was a smaller ovarian volume. At the same time, the ends of the ovary were cemented into a group in an “S” shape and not easily separated (Figure 2A–C,K). With increasing age, the length of ovarioles also increased slightly (Figure 2D–F,K).

The mean ovary size increased with maternal age, and this increase was linear (Figure 2K). Females exposed to high temperature exhibited a significant reduction in ovarian development time. On day 3, the length of the ovary under the 36 °C treatment reached the maximum value (0.89 mm) and then declined. On day 12, the ovary size was still larger than that of the newly emerged ovaries. The ovary length of the control peaked at 5 days (0.75 mm) and ovaries were observed for 12 days.

### 3.3. Effects of High Temperature on Egg Production

In addition to ovary size, the oocyte number also increased with temperature (*F*_(1,298)_ = 29.89, *p* < 0.05). In any case, few or even no mature eggs were detected upon new emergence (Figure 2A,B). After emergence, the oocytes progressively developed into mature oocytes, and the number of mature oocytes in the ovary gradually increased (Figure 2D–F,L). Most oocytes were observed on day 3 for the heat treated females (*n* = 24.07) (Figure 2D,L)and on day 5 for the controls (*n* = 12.06) (Figure 2E,L). In addition, mature oocytes had oocyte stalk hook analogues (Figure 2C–J). Then, the number of mature eggs in the ovaries declined (Figure 2G–J). At the end of ovarian development, mature oocytes gradually disappeared, but the structure similar to oocyte stalk hook analogues did not (Figure 2I,J,M).

### 3.4. Effects of High Temperature on Testicular Development

The reproductive system of male *A. bambawalei* is composed of a pair of testes (ts), vas deferens (vd), seminal vesicles (sv), accessory glands and a single ejaculatory duct (ed), and penis (Figure 3A–H). White translucent testes are located in front of the reproductive organs. The testes were spindle-shaped, but their sizes varied at different stages (Figure 3A–H,I,J). Specifically, in the early stages of emergence, the testes were filled with bundles of sperm (Figure 3A,B,I,J), while, after a period of emergence, the testes decreased in size and flattened due to the large number of sperm released into the seminal vesicle (Figure 3C–H,K,L).

The short vas deferens (vd) was connected to the base of the testis, and the lower part of the vas deferens was enlarged into a yellow seminal vesicle (sv). On days 3 to 5 after emergence, the seminal vesicle was filled with mature sperm (Figure 3C–F).

High temperature had a strong impact on the development of testes (Figure 3I,J). Over time, the length of testes at 36 °C was significantly longer than that of the control (*F*_(1,418)_ = 194.74, *p* < 0.05) (Figure 3I). However, no changes were observed in the width of testes (*F*_(1,418)_ = 1.1813, *p* = 0.278) (Figure 3J). Overall, testis size (including length and width) tended to decrease with male age (length: *F*_(1,418)_ = 35.717, *p* < 0.05; width: *F*_(1,418)_ = 103.58, *p* < 0.05).

### 3.5. Effects of High Temperature on Seminal Vesicle Development

At all ages, males at high temperature had larger seminal vesicles than control males (Figure 3K,L) (length: *F*_(1,418)_ = 69.239, *p* < 0.05; width: *F*_(1,418)_ = 29.735, *p* < 0.05). In heat-treated and control males, the seminal vesicle size increased with age and then decreased. On day 3, the seminal vesicle size under the 36 °C treatment reached the maximum value (0.126 × 0.071 mm), while the seminal vesicle size of the control peaked on day 4 (0.109 × 0.066 mm) (length: *F*_(1,418)_ = 35.717, *p* < 0.05; width: *F*_(1,418)_ = 103.58, *p* < 0.05).

## 4. Discussion

High temperature accelerates the introduction, colonization, population establishment, and spread of biological invasions [3], and can also affect the migration of accompanying natural enemies [2]. To control invasive insects, the accompanying natural enemies must positively respond to the climatic conditions of the invaded area [4,5,6]. To clarify whether high temperature will shorten the development period or enhance the development of the reproductive systems of the predominant accompanying natural enemies of *P. solenopsis*, we compared the pupal development, hind tibia of female adults, and fecundity of *A. bambawalei* under different temperatures.

First, high temperature accelerates the growth and development rate of insects [3]. As expected, in our study, the temperature in the growth chamber was negatively correlated with pupal developmental duration and positively correlated with fecundity. The results showed that, when the temperature in the growth chamber was raised from 27 °C to 36 °C, the pupal developmental duration of *A. bambawalei* was significantly shortened from 7.05 to 5.63 days, and the fecundity and growth rate increased by 40%. The above results are consistent with those of Yi et al. [30]; that is, at higher temperatures, rapid ovarian maturation and acceleration of egg and embryonic development of insects were observed. These results indicate that high temperatures may be conducive to *A. bambawalei* [31].

Of course, only the effects of high temperature on the pupal development duration and gonad development of *A. bambawalei* were studied, and the experiment was only carried out for a short time. Thus, it was difficult to clarify the biological response of this parasitoid to rapid climate change. Therefore, we also studied the effects of high temperature on other aspects of this parasitoid. The size of many parasitoids is generally measured by the length of the hind tibia or forewing [27]. In our study, the hind tibia of *A. bambawalei* female adults were also affected by temperature, and the length of the hind tibia at 36 °C was 1.16 times that of the control. On the other hand, the number of mature eggs in the ovary of the female adult wasp is directly related to the individual size of the wasp [26]. Our results were consistent with this previous observation; that is, high temperature resulted in a larger female wasp and a greater oviposition potential. These results indicated that insects that migrate from tropical zones [32] invest more in reproduction [33]. Regardless, as a result, there will be greater overwintering survival, higher reproductive rates, and an increased number of generations of these insects [34]. These trait changes are conducive to increasing spread in new areas. In contrast, previous studies have pointed out that increased temperature may led to a smaller size [35]. This divergence may be caused by the phenotypic plasticity of different organisms [36]. In this study, 36 °C treatment resulted in the increased size of *A. bambawalei*, which may be related to its strong phenotypic plasticity.

At present, compared with more reports focusing on the impact of high temperature on the distribution [3,13,23,32], survival [15], development [16,17], and reproduction [17,24] of *P. solenopsis* and its parasitoid, there are relatively few reports on morphological characteristics. Phenotype is the result of the interaction of genetic factors and environmental factors (biological and abiotic). *P. solenopsis* originated in North America and then invaded many countries, including those in South America, Africa, and Asia [3]. The genetic analysis of the North American and Asian groups of *P. solenopsis* shows that there has been great differentiation between the two groups [22]. At the same time, the average temperature in China showed an increasing trend from 1961 to 2018, and the increasing trend rate in most areas exceeded 0.8 °C/10a [37]. Therefore, in the future, we need to verify whether the genotype of *A. bambawalei* has changed in different regions to speculate whether this parasitoid is adapting to global warming.

To determine the egg quantity more accurately, we deprived the host of *A. bambawalei* according to other studies. Unexpectedly, some oocyte stalk hook analogues were produced during egg maturation. After the egg disappeared, this structure persisted in the ovary. A previous study demonstrated that oocytes remain attached to the ovary by a cellular pedicel until the oocytes are released into the ovary lumen [38]. However, in this study, oocyte stalk hook analogues only occurred in mature oocytes. Therefore, the egg pedicel may not be the key structure involved, and the function of this structure needs to be investigated further. Meanwhile, this structure is a part of the mature egg and does not disappear after egg resorption. Therefore, we can estimate the quantity of mature eggs from this structure. The more mature eggs there are at high temperature, the more oocyte stalk hook analogues are produced after egg resorption. However, the effect of high temperature on egg resorption remains unclear. On the other hand, egg resorption of female parasitoids occurs under host deprivation [39]. This is often attributed to insufficient nutrition of parasitoids [40] or the reproductive and developmental strategy of parasitoids [41]. In this study, we provided 10% honeydew solution to *A. bambawalei* and renewed it every day until dissection. This nutrient source has been proven to provide parasitoids with the necessary nutrients [24]. Moreover, the number of oocyte stalk hook analogues was lower when the host was properly provided. Therefore, we can speculate that egg resorption (the generation of oocyte stalk hook analogues) is mainly due to the reproductive and developmental strategy of parasitoids.

Finally, similar to other hymenopteran insects, the fertilized eggs of *A. bambawalei* develop into females, while unfertilized eggs develop into males [42]. For parasitoids, a female-biased population is more beneficial for their control capacity [43]. Thus, in addition to egg number and size, the quantity of sperm and the fertilization of parasitoids also influence the control capacity [29]. Parasitoid sperm are generated in the testis, and the vas deferens then transports mature sperm to the seminal vesicle [28]. Thus, sperm in the seminal vesicle are all mature sperm [29]. In a few studies, larger organs contained more sperm, and sperm stock was measured as the male reproductive capacity [44]. In the present study, we evaluated the quantity of mature sperm by measuring the size of seminal vesicles. The results indicated that high temperature accelerates sperm maturation and increases sperm quantity. The developmental time was shortened from 4 days to 3 days, and the seminal vesicle size increased from 0.109 × 0.066 mm to 0.126 × 0.071 mm. This also confirms our previous hypothesis that these temperature-related changes in reproductive morphology may influence the female offspring ratio increase in *A. bambawalei*. In contrast, high temperature may cause delays in spermatogenesis [29] or decreases in sperm motility [45], and is not conducive to the production of sperm of other parasitoids, thus reducing the proportion of female offspring. For example, in *Nasonia vitripennis*, the same heat treatment was proven to be harmful to sperm quantity but did not impair sperm quality [26]. To more precisely evaluate the impact of high temperature on sperm, the quantity and validity of mature sperm should be calculated further.

Since global warming does not produce an even impact on all ecological habitats, how global warming will affect parasitoids and their role as natural enemies of pests is difficult to evaluate in the short term. However, the results of this study are still helpful for understanding the adaptive mechanisms of *A. bambawalei*.

## 5. Conclusions

A study of the developmental, morphological, and fecundity changes in the parasitoid at high temperature therefore provides crucial insight into its response to temperature change [45,46]. In this study, appropriate high temperature was proven to shorten the pupal developmental duration, alter the hind tibia of female adults, and facilitate the fecundity of *A. bambawalei*. Therefore, this may explain, to some extent, why *A. bambawalei* can migrate with the mealybug, and why high temperature can increase the sex ratio of the offspring of *A. bambawalei*. Future studies should be carried out on the reproductive physiology, biochemistry, and molecules involved to clarify the specific accompanying mechanism. A clear definition of this mechanism will be conducive to the better application of *A. bambawalei* to prevent and control *P. solenopsis* and will enable the prevention and control of other biological invasions under the current situation of high temperature.

## Figures and Tables

**Figure 1 insects-12-00833-f001:**
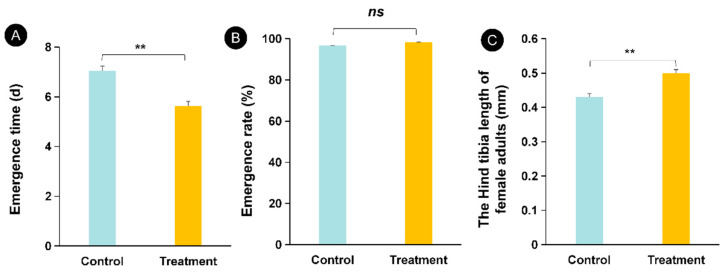
Effect of high temperature on pupal development (**A**) emergence rate (**B**) and the hind tibia of female adults (**C**) of *Aenasius bambawalei*. ** indicates that there is a significant difference between the high temperature treatment and the control.

**Figure 2 insects-12-00833-f002:**
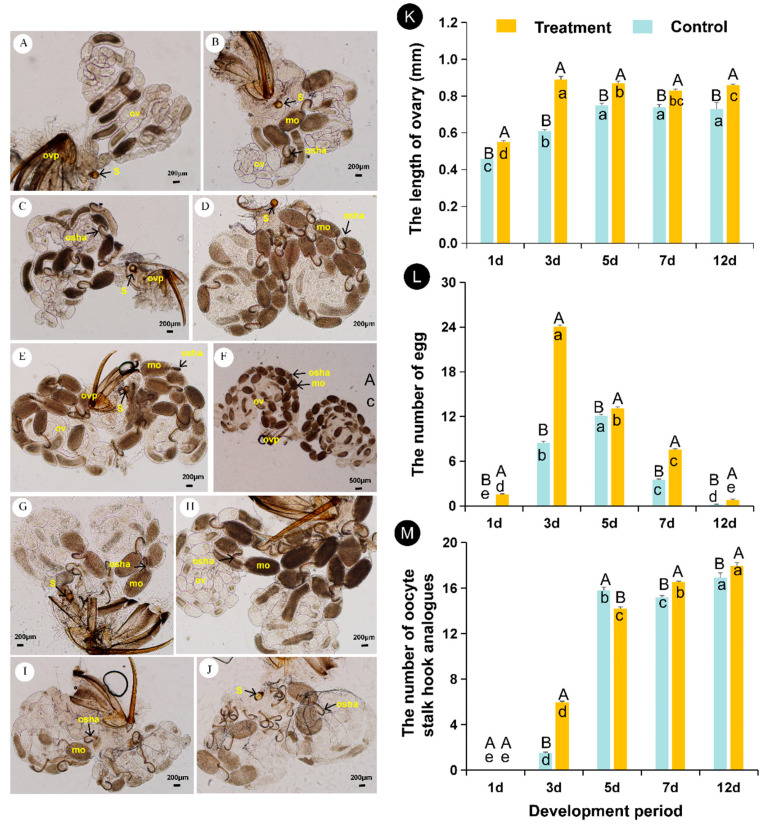
Morphology and developmental dynamics of *Aenasius bambawalei* ovaries at different temperatures. (**A**,**C**,**E**,**G**,**I**) represent the morphology of ovaries from control females at the early stage i (1 day after emergence), early stage ii (3 days after emergence), middle stage (5 days after emergence), final stage i (7 days after emergence), and final stage ii (12 days after emergence), respectively; (**B**,**D**,**F**,**H**,**J**) represent the morphology of ovaries from heat-treated females at the early stage i (1 day after emergence), early stage ii (3 days after emergence), middle stage (5 days after emergence), final stage i (7 days after emergence), and final stage ii (12 days after emergence), respectively. In (**K**,**L**,**M**), the different uppercase letters indicate that there is a significant difference between the high temperature treatment and the control, while the different lowercase letters indicate that there is a significant difference at different developmental periods. Ovp—ovipositor, ov—ovary, s—spermatheca, mo—mature oocytes, osha—oocyte stalk hook analogs. Bars in-(**A**–**J**) (except (**F**)) indicate 200 μm.

**Figure 3 insects-12-00833-f003:**
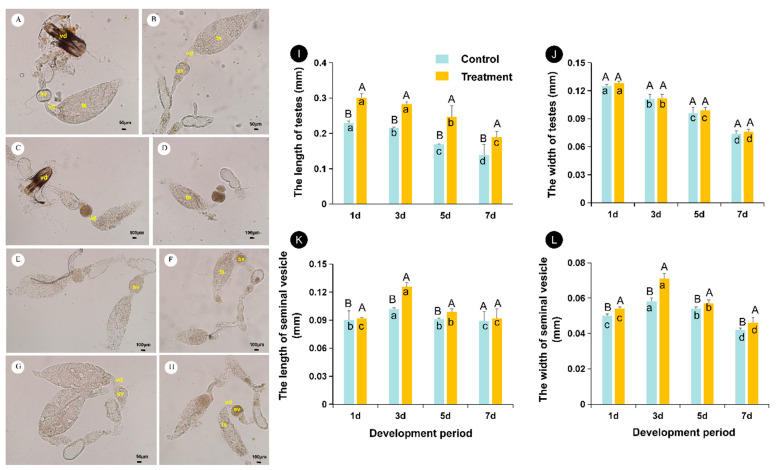
Morphology and developmental dynamics of *Aenasius bambawalei* testes at different temperatures. (**A**,**C**,**E**,**G**) represent the morphology of testes from control males at the early stage (1 day after emergence), middle stage i (3 days after emergence), middle stage ii (5 days after emergence), and final stage (7 days after emergence), respectively; (**B**,**D**,**F**,**H**) represent the morphology of testes from heat-treated males at the early stage (1 day after emergence), middle stage i (3 days after emergence), middle stage ii (5 days after emergence), and final stage (7 days after emergence), respectively. In (**I**–**L**), the different uppercase letters indicate that there is a significant difference between the and the high temperature treatment and the control, while the different lowercase letters indicate that there is a significant difference at different development periods. Ts—testis, vd—vas deferens, sv—seminal vesicle, ed—ejaculatory duct. Bars A–F, H indicate 0.1 mm, and bar G indicates 0.05 mm.

## Data Availability

The data of the research were deposited in the Zhejiang Institute of Landscape Plants and Flowers/Zhejiang Xiaoshan Institute of Cotton & Bast Fiber Crops.

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
