# Peer review of "The Effects of Temperature on the Development, Morphology, and Fecundity of Aenasius bambawalei (=Aenasius arizonensis)"

_insects, 2021, doi:10.3390/insects12090833_

Round 1

Reviewer 1 Report

Paper submitted by Zhang et al presents original experimental results on the reproductive tract of a parasitoid wasp reared at control temperature, and exposed to a heat period of 24hrs during their pupal development. The protocol resembles this of Chirault et al., PlosOne 2015 (cited) or Nguyen et al., 2013 J ins Physiol (not cited). The studied parasitoid is of major interest for the control of an insect pest spreading in cultures around the world. Authors ask the question of the adaptation of the reproduction of both females and males to high temperatures, in new invaded environments of the pest.

In general, the paper is concise and well presented; I suggest some adds to improve it.

Authors used the words “heat stress”, but for me a stress is a negative physiological response to an aggression, it should induces a decrease of fitness. Here, such called stress induced favourable changes. For me, it is not a stress but an environmental condition that is better than control. As a base for the relation between ectotherms and temperature, authors may refer to Huey, R.B. & Stevenson, R.D., 1979. Integrating thermal physiology and ecology of ectotherms: discussion of approaches, American Zoology, 19, 357-366. In the discussion, we are waiting for some comparisons with other wasps which were investigated for heat stress, leading to a decrease of fertility.

I appreciated the fine description of female and male reproductive apparatus. This could be completed by some comparisons with other wasps species in the discussion.

Considering female organs, especially ovaries, authors may discuss on the hypothesis of an acceleration of the development due to high temperature, which could lead to more mature females. This is reinforced by the shorter development time showed on figure 1. One question is do heat-exposed females and males live shorter than controls?

Considering female hind tibia length, I cannot understand what may provoke an increase, because the heat period occurs at the pupal stage, when appendices are already developed. Could authors give some information about the stage actually considered in the experiments.

In conclusions, some words about sex ratio would be welcome to link this work with the use of this wasp in biocontrol.

Minor points:

Figure 3 is too small and not sufficiently contrasted for pictures. Authors must write that males are virgin, because sperm transfer could change the shape of testis and seminal vesicles.

Line 113, sentence could be “…microscope to detect intact or impaired female…”

Author Response

Dear reviewer:

We are very grateful to your comments for the manuscript. According with your advice, we amended the relevant part in manuscript. The questions were answered below.

Point 1: Authors used the words “heat stress”, but for me a stress is a negative physiological response to an aggression, it should induces a decrease of fitness. Here, such called stress induced favourable changes. For me, it is not a stress but an environmental condition that is better than control. As a base for the relation between ectotherms and temperature, authors may refer to Huey, R.B. & Stevenson, R.D., 1979. Integrating thermal physiology and ecology of ectotherms: discussion of approaches, American Zoology, 19, 357-366. In the discussion, we are waiting for some comparisons with other wasps which were investigated for heat stress, leading to a decrease of fertility.

Response 1: It is really true as reviewer suggested that in our manuscript “high temperature” is not a stress but an environmental condition that is better than control. 

In view of this, we carefully read the papers recommended by the reviewers and cited them in appropriate places in our papers(Line 58 ï¼‰

In addition, we have added some comparisons with other wasps which were investigated for heat stress, leading to a decrease of fertility.(Line 305-308:  Such as in...)

Point 2: I appreciated the fine description of female and male reproductive apparatus. This could be completed by some comparisons with other wasps species in the discussion.

Response 2: Considering the Reviewer's suggestion, we have added some comparisons with other parasitoid species in the discussion as the following: ï¼ˆLine 302-305: By contrast, ...)

Point 3: Considering female organs, especially ovaries, authors may discuss on the hypothesis of an acceleration of the development due to high temperature, which could lead to more mature females. This is reinforced by the shorter development time showed on figure 1. One question is do heat-exposed females and males live shorter than controls?

Response 3: Thank you for pointing this out. Indeed, the longevity of some parasitoids  decrease with the increase of temperature. However, although not explicitly stated, high temperature treatment did not have a significant effect on the longevity of A. bambawalei in this study. 

Point 4: Considering female hind tibia length, I cannot understand what may provoke an increase, because the heat period occurs at the pupal stage, when appendices are already developed. Could authors give some information about the stage actually considered in the experiments.

Response 4: As reviewer suggested that female hind tibia begin to develop at a specific period, and we have consulted the relevant literature. At present, there is a lack of literature on the specific development period of the appendages of the parasitoid. Our initial aim was to select a temperature that might cause the greatest difference. According to the previous studies and our previous research results, we found that 36 â„ƒ for 24 hours could cause a great change in the sex ratio of the offspring of the wasp. At the same time, we unexpectedly found that the pupal stage of high temperature treatment will increase the posterior tibia, but the mechanism is not clear.

Point 5: In conclusions, some words about sex ratio would be welcome to link this work with the use of this wasp in biocontrol.

Response 5: We are grateful for the suggestion. In fact, this study is based on our previous study that high temperature can increase the sex ratio of the offspring of A.bambawalei.To be more clearly and in accordance with the reviewer concerns, we have added a more detailed in the revision (Line 315-316: and why high temperature can increase the sex ratio of the offspring of A.bambawalei. )

Point 6: Authors must write that males are virgin, because sperm transfer could change the shape of testis and seminal vesicles.

Response 6: We agree with the comment and re-wrote the sentence in the revised manuscript as the following:  (Line115: 30 virgin females or virgin males per treatment group)

Point 7: Line 113, sentence could be “…microscope to detect intact or impaired female…”

Response 7:  Line , the statements of "microscope to determine intact female or male gonads" were corrected as "microscope to detect intact or impaired female or male gonads". (Line 119)

Reviewer 2 Report

This manuscript reports the effects of high temperature on pupal development, the hind tibia of female adults, ovarian development, egg production, testicular development and seminal vesicle development. The authors provided the biological performance of Aenasius bambawalei response to high temperature. This study is well organized and meaningful with applying global warming for the natural enemy. However, the authors explain reasonable information of materials and methods, results and discussions.

Major

  1. Only one high temperature is enough to explain the adaptation of bambawalei to high temperature?
  2. The authors give clear statistical information of results.
  3. References is not enough to support the introduction and hypothesis.
  4. The resolution of figures is not clear to understand the morphology and development of female and male sexual systems.
  5. The definition of global warming is so broad. Therefore, the authors carefully approach to clarify the effect of high temperature on bambawalei.
  6. I hope the authors give more discussions about the synchronization between bambawalei and its host.

Minor

Line 70-72. It is needed more logical explanation. 

Line 103. newly emergent? It means within 12hours or 24hours?

Line 107-130. Please give more precise description of experimental process and materials.

 Line 111: place in the freezer? à Please give the information of temperature.

 Line 113: Please give the general information of anatomical microscope.

 Line 115: Please give the general information of the digital camera.

 Line 119-120. How did you choose the largest mature oocytes?

Line 142-202. Please give the information of statistical analysis.

Line 183. Figure 2 II A à Figure 2 II B.

Line 225-231. Please give appropriate and more references.

Figure 2 II B. Would you check the lowercase letters “d” and “e”? The value of “d” and “e” are “0”?

Figure 3 II C. Would you check the uppercase letters?

Author Response

Dear reviewer:

Thanks a lot for your comments for the manuscript. Those comments are all valuable and very helpful for the revising and improving our manuscript, as well as the important guiding significance to our researches. We have studied comments carefully and have made correction which we hope meet with approval. And the main correction in the paper and the responds to the reviewer’s comments are as following:

Point 1: Only one high temperature is enough to explain the adaptation of A.bambawalei to high temperature?

Response 1: Special thanks to you for your good comments. Indeed, adaptation is a long-term process. After careful consideration, we think it would be more appropriate to express the “adaptation” as “responses and alteration” at high temperature, and the part involving adaptation in the full text has been modified.

In addition, the temperature setting and insect stage selection in this study are based on our previous study: treating the pupal stage at 36 â„ƒ for 24 h can increase the proportion of females offspring of A. bambawalei. 

In short, one high temperature ( 36 â„ƒ exposed for 24 h) could alter the development, morphology, and fecundity of A.bambawalei. 

Point 2: The authors give clear statistical information of results. 

Response 2: Considering the reviewer's suggestion, on the basis of increasing the statistical method of emergence rate and the  hind tibia length, we also modified and improved other parts of statistical analyses.(Line 141-143: For the emergence rate, it is the percentage of the number of emerged A. bambawalei in the number of parasitized P. solenopsis. The effect of...)

Point 3: References is not enough to support the introduction and hypothesis. 

Response 3: Thank you for pointing this out. In the revision, we revised the statements of "global warming" and "adaptation", and compared the results of this study with the reproductive characteristics of other parasitoids at high temperature. At the same time, corresponding references have also been added to the revised draft.(Line 360-361, Line 402-403)

Point 4: The resolution fo figures is not clear to understand the morphology and development of female and male sexual systems.

Response 4: We are very sorry that our figures resolution are low, which make it difficult to read. Thus, we have re-adjusted the resolution of the figures to make them look clearer to the reviewer's suggestion. (Line 175, Line 211)

Point 5: The definition of global warming is so broad. Therefore, the authors carefully approach to clarify the effect of high temperature on A. bambawalei.

Response 5: As Reviewer suggested that the definition of global warming is so broad. In fact, in this experiment, we put more emphasis on the effect of high temperature on A. bambawalei. Therefore, in the revision, we evaded the statement of “global warming” and directly changed to “high temperature”.

Point 6: I hope the authors give more discussions about the synchronization between A. bambawalei and its host.

Response 6: Thank you for pointing this out. Previous studies have shown that A. bambawalei was recorded with P. solenopsis at all sites surveyed in Pakistan, India, and China, and there will be a congruent spread of this mealybug and its parasitoids across China.

Therefore, parasitoids and mealybugs are more synchronous in space, and there is little report on temporal synchronization at present. And we have added this details in the revision. (Line 261-262)

Point 7: Line  70-72. It is needed more logical explanation. 

Response 7: We have made correction according to the reviewer's comments, and more details in the second paragraph of Introduction. (Line 73-75:Thus, we hypothesized that high temperature may cause some responses and alteration in development, morphology and fecundity of A. bambawalei.)

Point 8: Line  103. newly emergent? It means within 12hours or 24hours?

Response 8:  I'm sorry we didn't make it clear. It means within 12hours (Line 113: Newly emergent (within 12 h) ).

Point 9: Line 107-130. Please give more precise description of experimental process and materials.

Response 9:  Considering the reviewer's suggestion, we have added more precise description of experimental process and materials in our revision, and more details in the materials and methods. (Line 84-92).

Point 10: Line 111: place in the freezer? à Please give the information of temperature.

Response 10:  Thank you for point it out. We have added the information of temperature in the revision. (Line 117: placed in the freezer (average temperature, -20 °C).

Point 11: Line 113: Please give the general information of anatomical microscope.

Response 11:  Thanks a lot, the more details were added in the revision. (Line 119:anatomical microscope (Nikon SMZ-10) )

Point 12: Line 115: Please give the general information of the digital camera.

Response 12:  Indeed, as suggested by the reviewers, adding information from digital camera will make the manuscript more precise. Thus, we added the details. (Line 121: Canon D60 digital camera)

Point 13: Line 119-120. How did you choose the largest mature oocytes?

Response 13: Thanks for point it out. We added these details. (Line 126-127: The diameter of all mature oocytes were measured, and each of the 3 largest mature oocytes was selected as the indicator (mm). )

Point 14: Line 142-202. Please give the information of statistical analysis.

Response 14:  It is really true as Reviewer suggested that  the statistical method of emergence rate and the  hind tibia length and some modification on other parts of statistical analyses are needed, and we added these details in the revision.(Line 141-143)

Point 15: Line 183. Figure 2 II A à Figure 2 II B.

Response 15: Considering the reviewer’s suggestion, we  have re-adjusted the resolution of the Figure 2 II A à Figure 2 II B to make them look clearer to the reviewer's suggestion. (Line 175)

Point 16: Line 225-231. Please give appropriate and more references.

Response 16: Just as reviewer advised, adding more appropriate references will be conducive to the expression of manuscript, thus we added in this part. (Line 360-361, Line 402-403)

Point 17: Figure 2II B. Would you check the lowercase letters “d” and “e”? The value of “d” and “e” are “0”?

Response 17: Thank you very much for your reminding. We carefully checked the values of D and e again. Because the difference between the data is relatively small, the standard error is about 0.03, which is quite different from the average value. Thus, it is not shown in the picture and is close to not obvious.

Point 18: Figure 3II C. Would you check the uppercase letters?

Response 18: Just as the point 17, we have checked the uppercase letters and made corresponding changes in the revision.

Point 19: The problem I have with the manuscript is its linkage to global warming.  If the linkage to global warming is minimized the manuscript could be salvaged.

Response 19:   It is really true as Reviewer suggested that the definition of global warming is too broad. In fact, in this experiment, we put more emphasis on the effect of high temperature on A. bambawalei. Therefore, in the revision, we evaded the statement of “global warming” and directly changed to “high temperature”.

Reviewer 3 Report

The problem I have with the manuscript is its linkage to global warming.  If the linkage to global warming is minimized the manuscript could be salvaged.

Manuscript Review – The effects of temperature on the development, morphology, and fecundity of Aenasius bambawalei (=Aenasius arizonensis): is it adapted to high temperature?

General remarks

The methodology of this investigation is standard.  The results are well-stated and are more or less as expected.  The general problem with the study is the conceptual framework, particularly with too much reference to global warming.  

It is already known that insects are exothermic invertebrates in which activities and development are largely dependent on ambient temperatures.  The results obtained in this study are largely in agreement of most studies where insects are studied under different environmental temperatures. An increase in temperature accelerates development leading to early maturity and with adequate nutrients larger body sizes that may result in larger broods, until, at least, the optimum temperature is exceeded. 

Global warming is a long-term effect that its impact must be deduced by observing similar events or responses over a space of time.  If the responses of Aenasius bambawalei to high temperature has changed over several years due to the warming of the environment that can be ascribed to global warming.  Is it not possible that if the current experiment was carried out 50 years at the same temperatures you would have obtained the same results?  You can also demonstrate a gradual shift in the responses of A. bambawalei to high temperatures by comparing your results to earlier data obtained by other authors (if available).

My suggestion will be to modify your strong linkage of your work to global warming by focusing on the effects of elevated temperatures on A. bambawalei, while in your discussion speculate on possible impact of global warming on this parasitoid.

Specific comments

Title- The effects of temperature on --------------------------------------------------: Adaptive features?

Summary

Ln 17: Not easily understood – predominantly associated this mealybug?

Ln 23:  This is still queried?  All insects respond to elevated temperature.  Like I said, there is no evidence to show that these insects would have responded differently if this experiment was carried out several years ago.

Abstract

Ln 24: Should read the effect of elevated temperatures on ---------------------

Ln 29-31: This experiment is not enough to conclude that the change is an adaptation of somewhat.

Ln 39-40: You did not elucidate on this anywhere in the manuscript.
Introduction

Ln 62: delete been

Ln 66: However, in some areas newly invaded by P. solenopsis ----------

Ln 66-68: Why? Colder or what?

Ln 73: Hypothesis should be higher temperatures could enhance development and size of the parasitoid. 

Ln 85: Controlled conditions -which were?

Ln 88: Change rejuvenate -to maintain genetic stability and fitness of the populations.

Ln 97: Climate room??

Ln 103: Delete “counted and”

Ln 103-105: Any reference on the use of tibia length to determine fecundity? 

Ln 116: Was the digital camera coupled to a microscope?

Ln 123-128: How were the oocytes and sperm cells observed? Was the Nikon SMZ1500 used as well? 

Results

Figs: How was emergent rate evaluated or was it just percentage emergence?

Ln 156: Were all the oocytes counted or did you count only mature (chorionated) oocytes?

Fig. 2. You may wish to change ‘amount’ to ’number’.  Number is countable while amount may be estimated as volume.

Ln 186-189: This section lacks clarity.  Are you saying that oocyte maturation declined or that the number of mature oocytes in the ovaries declined?  If the later was the case are you suggesting egg resorption?

Ln 192: Delete ‘In any case’.

Ln 212-231: The increase in the size of male reproductive system was it accompanied by increase in sperm cells?

Discussion

It is generally known that high temperatures accelerate development in most exothermic animals including insects.  What is not fully understood is how global warming will impact or interface with general development of arthropods especially since global warming does not produce an even impact on all ecological habitats.  Sometimes undue cold streams have been attributed to global warming.  Perhaps, adaptation to global warming can be assessed by comparing the responses of the same organisms over time, say 20 years.  A shift in those responses could be described as adaptation. 

The discussion should be revised in the light of this comment.

Author Response

Dear reviewer:

Thanks very much for taking your time to review this manuscript. We really appreciate all your generous comments and suggestions. According to your advice, we amended the relevant part in manuscript. All of your questions were answered one by one.

Point 1: Title- The effects of temperature on --------------------------------------------------: Adaptive features?

Response 1: We are grateful for the suggestion. Indeed, adaptation is a long-term process. After careful consideration, we think it would be more appropriate to express the “adaptation” as “responses and alteration” at high temperature, and  we have modified this expression throughout the text according to the comment.

Thus, we corrected the title to: The effects of temperature on the development, morphology, and fecundity of Aenasius bambawalei (=Aenasius arizonensis), an accompanying parasitoid of Phenacoccus solenopsis.

Point 2: Ln 17: Not easily understood – predominantly associated this mealybug?

Response 2: We are very sorry for your misunderstanding. “predominant accompanying parasitoid” actually includes two meanings, that is, A. bambawalei is the most effective natural enemy of P. solenopsis, and this parasitoid can also migrate with mealybug.

Point 3: Ln 23:  This is still queried?  All insects respond to elevated temperature.  Like I said, there is no evidence to show that these insects would have responded differently if this experiment was carried out several years ago.

Response 3: Thank you for pointing this out. Indeed, all insects respond to elevated temperature. In fact, we mainly want to emphasize the relationship between this result and previous research results. Therefore, we have made the modifications to this part as following: Our results indicated that moderately high temperature was conducive to the reproduction and development of the parasitoid, and may be related to the more females offspring. ï¼ˆLine 21-22)

Point 4: Ln 24: Should read the effect of elevated temperatures on ---------------------

Response 4: We deeply appreciate the reviewer’s suggestion. According to the reviewer’s comment, we have revised this part as follows: The effects of high temperature on the developmental and morphological characteristics of insects, including biological invaders and their accompanying natural enemies, are obvious. (Line 25-27)

Point 5: Ln 29-31: This experiment is not enough to conclude that the change is an adaptation of somewhat.

Response 5: Thank you for your comment, and just as you pointed out: this experiment is not enough to conclude that the change is an adaptation of somewhat. Indeed, adaptation is a long-term process. After careful consideration, we think it would be more appropriate to express the “adaptation” as “responses and alteration” at high temperature. And in this part, our reply is as follows: However, whether this increase is a result of enhanced development, morphology and fecundity of A. bambawalei caused by high-temperature remains unknown. (Line 30-32)

Point 6: Ln 39-40: You did not elucidate on this anywhere in the manuscript.

Response 6:  We are extremely grateful to reviewer for pointing out this problem. Just in point 5, adaptation is not suitable, thus we revised it as follows: These findings will help to clarify the response and alteration mechanisms of accompanying natural enemies.  (Line 40-41)

Point 7: Ln 62: delete been

Response 7:  Thank for you point it out, and “been”has been deleted. (Line 65)

Point 8: Ln 66: However, in some areas newly invaded by P. solenopsis ----------

Response 8:  Line 69-70, the statements of "However, in some newly invaded areas of P. solenopsis" were corrected as "However, in some  colder areas where newly invaded by P. solenopsis".

Point 9: Ln 66-68: Why? Colder or what?

Response 9:  I'm very sorry, but we have not stated clearly. And in the revision, we corrected it as follows: However, in some colder areas where newly invaded by P. Solenopsis. (Line 69-70)

Point 10: Ln 73: Hypothesis should be higher temperatures could enhance development and size of the parasitoid. 

Response 10:  Special thanks to you for your good comments. We reply this part as follows: Thus, we hypothesized that high temperature may cause some responses and alteration in development, morphology and fecundity of A. bambawalei. (Line 73-75)

Point 11: Ln 85: Controlled conditions -which were?

Response 11: Considering the reviewer's suggestion, we have added some details about this part: P. solenopsis was reared on cotton plants in a growth chamber (27–30 °C, 50–70% RH, and an L16:D8 photoperiod). (Line 86-87) 

Point 12: Ln 88: Change rejuvenate -to maintain genetic stability and fitness of the populations.

Response 12:  Line 92, the statements of "rejuvenate" were corrected as "to maintain genetic stability and fitness of the populations".

Point 13: Ln 97: Climate room??

Response 13:Thank you for pointing this out. In order to make it clear, we have revised it as  “then maintained in the growth chamber”. (Line 101)

Point 14:Ln 103: Delete “counted and”

Response 14: According to the reviewer's suggestion, “counted and” have been deleted. (Line 107)

Point 15: Ln 103-105: Any reference on the use of tibia length to determine fecundity? 

Response 15: We agree with the comment and re-selected a reference related to the use of hind tibia length to determine fecundity as follows: Sagarra, L. A.; Vincent, C.; Stewart, R. K . Body size as an indicator of parasitoid quality in male and female Anagyrus kamali (Hymenoptera: Encyrtidae). B. Entomol. Res. 2001, 91, 363-367. (Line 402-403)

Point 16: Ln 116: Was the digital camera coupled to a microscope?

Response 16: Thank you for point this out. Indeed, the digital camera coupled to a microscope, and we corrected it in the revision (Line 121).

Point 17: Ln 123-128: How were the oocytes and sperm cells observed? Was the Nikon SMZ1500 used as well? 

Response 17:  Thanks for your reminder. We corrected these two parts in the revision: These measurements were proceeded under a Nikon SMZ1500 zoom stereomicroscope (Nikon, Japan) and used to determine the mean oocyte size per female. (Line 128-129)

…the size of the seminal vesicle was measured under a Nikon SMZ1500 zoom stereomicroscope (Nikon, Japan) …(Line 131-132)

Point 18: Figs: How was emergent rate evaluated or was it just percentage emergence?

Response 18: For the emergence rate, the number of parasitized P. solenopsis was counted, and then the number of parasitic wasps emerging was counted (Line 106-107), and at last the percentage of the number of emerged wasps in the number of parasitized P. solenopsis, that is the emergence rate. This part of the statistical analysis method has been supplemented in the statistical analyses.(Line 141-143)

Point 19: Ln 156: Were all the oocytes counted or did you count only mature (chorionated) oocytes?

Response 19:  Thanks for point this out, and we counted only mature oocytes.

Point 20:  Fig. 2. You may wish to change ‘amount’ to ’number’.  Number is countable while amount may be estimated as volume.

Response 20:  Line , the statement of "amount" was corrected as "number". (Line 175)

Point 21: Ln 186-189: This section lacks clarity.  Are you saying that oocyte maturation declined or that the number of mature oocytes in the ovaries declined?  If the later was the case are you suggesting egg resorption?

Response 21:  I'm very sorry, but we have not stated clearly. What we want to say is the number of mature oocytes in the ovaries declined, and it is the case we suggesting egg resorption. In order to make it more clear, we revised this part as follows: Then, the number of mature eggs in the ovaries declined. (Line 196)

Point 22: Ln 192: Delete ‘In any case’.

Response 22: “In any case” have been deleted. (Line 163, Line 200)

Point 23: Ln 212-231: The increase in the size of male reproductive system was it accompanied by increase in sperm cells?

Response 23: Thank you for point this out. The reproductive system of male A. bambawalei is composed of a pair of testes (ts), vas deferens (vd), seminal vesicles (sv), accessory glands and a single ejaculatory duct (ed) and penis (Figure 3â… ).  (Line 200-202).

As larger organs contain more sperm, and the sperm in the seminal vesicle are mature, the size of the seminal vesicle was measured under a Nikon SMZ1500 zoom stereomicroscope (Nikon, Japan) to evaluate testis development. (Line 130-132)

Point 24: It is generally known that high temperatures accelerate development in most exothermic animals including insects.  What is not fully understood is how global warming will impact or interface with general development of arthropods especially since global warming does not produce an even impact on all ecological habitats.  Sometimes undue cold streams have been attributed to global warming.  Perhaps, adaptation to global warming can be assessed by comparing the responses of the same organisms over time, say 20 years.  A shift in those responses could be described as adaptation. 

The discussion should be revised in the light of this comment.

Response 24:  Special thanks to you for your good comments. Indeed, just as the reviewer pointed out, global warming does not produce an even impact on all ecological habitats. Sometimes under cold streams have been attributed to golbal warming. Actually, the temperature setting and insect stage selection in this study are based on our previous study: treating the pupal stage at 36 â„ƒ for 24 h can increase the proportion of females offspring of A. bambawalei. 

In addition, adaptation is a long-term process. After careful consideration, we think it would be more appropriate to express the “adaptation” as “responses and alteration” at high temperature, and the part involving global warming and adaptation in the full text have been modified.

Round 2

Reviewer 2 Report

I appreciate the authors’ hard works. The authors present the information of statistical analysis of effects of high temperature on testicular development in line 221-223 and 230. I hope the authors give appropriate statistical information of effects of high temperature on pupal development, hind tibia of female adults, and ovarian development.

Author Response

I appreciate the authors’ hard works. The authors present the information of statistical analysis of effects of high temperature on testicular development in line 221-223 and 230. I hope the authors give appropriate statistical information of effects of high temperature on pupal development, hind tibia of female adults, and ovarian development. 

Dear reviewer:

We are very grateful to your comments for the manuscript. According with your advice, we amended the relevant part in manuscript. The questions were answered below.

Point 1: The authors present the information of statistical analysis of effects of high temperature on testicular development in line 221-223 and 230. I hope the authors give appropriate statistical information of effects of high temperature on pupal development, hind tibia of female adults, and ovarian development. 

Response 1: As the reviewer pointed out, it would be more appropriate if we added the statistical information of effects of high temperature on pupal development, hind tibia of female adults, and ovarian.

In view of this, we have added this part in the revision. (Line 157-159: High temperature could significantly affect the pupal developmental duration (emergence time) (F(1,118)=13.50,P<0.05) (Figure 1A) and the hind tibia length of female adults (F(1,58)=22.63,P<0.05) (Figure 1C) but had no effect on the emergence rate (F(1,8)=0.40,P=0.54) (Figure 1B). ) 

Reviewer 3 Report

Extensive revision of the Introduction and Materials and Methods is required.  

Remarks

The general problem with the study is the conceptual framework, particularly with too much reference to global warming.  This problem was not addressed in the introduction. 

It is already known that insects are exothermic invertebrates in which activities and development are largely dependent on ambient temperatures. The results obtained in this study are largely in agreement of most studies where insects are studied under different environmental temperatures. An increase in temperature accelerates development leading to early maturity and with adequate nutrients larger body sizes that may result in larger broods, until, at least, the optimum temperature is exceeded.

Global warming is a long-term effect that its impact must be deduced by observing similar events or responses over a space of time.  If the responses of Aenasius bambawalei to high temperature has changed over several years due to the warming of the environment that can be ascribed to global warming.  Is it not possible that if the current experiment was carried out 50 years at the same temperatures you would have obtained the same results? You can also demonstrate a gradual shift in the responses of A. bambawalei to high temperatures by comparing your results to earlier data obtained by other authors (if available).

My suggestion will be to modify your strong linkage of your work to global warming by focusing on the effects of elevated temperatures on A. bambawalei, while in your discussion speculate on possible impact of global warming on this parasitoid.

Specific comments

Summary and Abstract

I am very comfortable with the phrase “responses and alteration”, which is found in several places in the summary and Abstract.  He phrase does not convey the meaning of the changes that were observed.  If the changes were shortening of developing period, or enhanced development of the reproductive systems, or if high temperature induced egg resorption, these should be stated clearly.

Introduction

The introduction requires major revision.  There is still great emphasis on climate change.  The premise for this study should laid squarely where it should be, that is the impact of temperature on development.  Climate change can be discussed with respect to the implications of the results under Discussion.

Results

Ln 167-193: Were all the oocytes counted or did you count only mature (chorionated – implies eggshell) oocytes? This needs to be clarified in the text.

Discussion

It is generally known that high temperatures accelerate development in most exothermic animals including insects.  What is not fully understood is how global warming will impact or interface with general development of arthropods especially since global warming does not produce an even impact on all ecological habitats.  Sometimes undue cold streams have been attributed to global warming.  Perhaps, adaptation to global warming can be assessed by comparing the responses of the same organisms over time, say 20 years.  A shift in those responses could be described as adaptation.

The discussion should be revised in the light of this comment. The revision of the discussion demanded in the last review was not done.  Only a few changes were inserted at intervals.

Ln: 235. Change to ---must positively respond

236: responses and alteration

241. Feeding temperatures? Was not mentioned in materials and methods. What does it mean?

271. Change ‘still existed” to persisted

276. after egg self-absorption – Change to egg resorption

Author Response

Extensive revision of the Introduction and Materials and Methods is required.  

Remarks

The general problem with the study is the conceptual framework, particularly with too much reference to global warming.  This problem was not addressed in the introduction. 

It is already known that insects are exothermic invertebrates in which activities and development are largely dependent on ambient temperatures. The results obtained in this study are largely in agreement of most studies where insects are studied under different environmental temperatures. An increase in temperature accelerates development leading to early maturity and with adequate nutrients larger body sizes that may result in larger broods, until, at least, the optimum temperature is exceeded.

Global warming is a long-term effect that its impact must be deduced by observing similar events or responses over a space of time.  If the responses of Aenasius bambawalei to high temperature has changed over several years due to the warming of the environment that can be ascribed to global warming.  Is it not possible that if the current experiment was carried out 50 years at the same temperatures you would have obtained the same results? You can also demonstrate a gradual shift in the responses of A. bambawalei to high temperatures by comparing your results to earlier data obtained by other authors (if available).

My suggestion will be to modify your strong linkage of your work to global warming by focusing on the effects of elevated temperatures on A. bambawalei, while in your discussion speculate on possible impact of global warming on this parasitoid. 

Dear reviewer:

Thanks a lot for your comments for the manuscript. Those comments are all valuable and very helpful for the revising and improving our manuscript, as well as the important guiding significance to our researches. In particular, the suggestions on adjusting global warming and high temperature adaptation will help to make our paper more rigorous. What we actually want to express is the impact of high temperature on accompanying natural enemies. Therefore, we have studied comments carefully and have modified the whole manuscript, including references which we hope meet with approval. And the main correction in the paper and the responds to the reviewer’s comments are as follows:

Point 1: Summary and Abstract 

I am very comfortable with the phrase “responses and alteration”, which is found in several places in the summary and Abstract.  He phrase does not convey the meaning of the changes that were observed.  If the changes were shortening of developing period, or enhanced development of the reproductive systems, or if high temperature induced egg resorption, these should be stated clearly.  

Response 1: Special thanks to you for your good comments. Indeed, just “responses and alteration” does not convey the meaning of the changes that were observed. Thus, in the summary and abstract, we clearly stated changes as “shorten the development period or enhance the development of the reproductive systems of the parasitoid” . 

Accordingly, we modify the content of the text. We replace “the responses and alteration of accompanying natural enemies to the environment, especially to temperature” with “whether temperature changes will shorten the development period or enhance the development of the reproductive systems of accompanying natural enemies” (Line 14-15)

In addition, we replaced “To understand the response and alteration of the parasitoid” with “To understand whether temperature changes will shorten the development period or enhance the development of the reproductive systems of the parasitoid” (Line 18-19)

In the end, we replaced “whether this increase is a result of responses and alteration of development, morphology and fecundity of A. bambawalei caused by” with “whether this increase is a result of shortening the development period or enhancing the development of the reproductive systems A. bambawalei caused by ” (Line 33-34) 

Point 2: Introduction

The introduction requires major revision.  There is still great emphasis on climate change.  The premise for this study should laid squarely where it should be, that is the impact of temperature on development.  Climate change can be discussed with respect to the implications of the results under Discussion. 

Response 2: Thank you for pointing this out. The suggestions on adjusting climate change will help to make our paper more rigorous. Actually, what we want to express is the impact of high temperature on accompanying natural enemies. Therefore, we have modified the whole manuscript (Line 49, 50, 56, 59, 60, in blue), including references (Line 341-344, in blue). 

Point 3: Results

Ln 167-193: Were all the oocytes counted or did you count only mature (chorionated – implies eggshell) oocytes? This needs to be clarified in the text. 

Response 3: Considering the reviewer's suggestion, we checked the results again and confirmed that we only counted the mature oocytes. In addition, we clarified it in the text. (Line 172, 174, 175, in blue)

Point 4: Discussion

It is generally known that high temperatures accelerate development in most exothermic animals including insects.  What is not fully understood is how global warming will impact or interface with general development of arthropods especially since global warming does not produce an even impact on all ecological habitats.  Sometimes undue cold streams have been attributed to global warming.  Perhaps, adaptation to global warming can be assessed by comparing the responses of the same organisms over time, say 20 years.  A shift in those responses could be described as adaptation. ()

The discussion should be revised in the light of this comment. The revision of the discussion demanded in the last review was not done.  Only a few changes were inserted at intervals.

Response 4: As Reviewer suggested that the definition of global warming is so broad. 

In fact, just the change of temperature is not enough to explain how global warming will impact or interface with general development of arthropods especially since global warming does not produce an even impact on all ecological habitats.

Moreover, we have only experimented for a period of time, which is not enough to show that this is an adaptive behavior of natural enemies.

Therefore, in the modified discussion, we defined the concepts of "high temperature", "shortening development duration", "improving reproductive efficiency" and so on. (Line 239, 241)

At the same time, in the references, we also replaced most of the literature related to "global warming" and "adaptability". (Line 267-269)

As for the description of specific results and comparison with other similar studies, we do not mention too much "global warming" or "adaptability". Therefore, not much modification has been made.

Point 5: Line 235. Change to ---must positively respond

Response 5: Thank you for pointing this out. “must positive responses” has been changed to “must positively respond”. (Line 238)

Point 6: Line 236: responses and alteration

Response 6:  I'm sorry we didn't make it clear. We have replaced “responses and alteration” with “To clarify whether high temperature will shorten the development period or enhance the development of the reproductive systems” (Line 239-240)

Point 7: Line 241. Feeding temperatures? Was not mentioned in materials and methods. What does it mean?

Response 7: We’re very sorry to trouble your understanding. What we want to express is the temperature of the insect room, which has been modified in the corresponding part of the article (Line 243, 245). This part was mentioned in materials and methods. (Line 86: in a growth chamber (27–30 °C, 50–70% RH, and an L16:D8 photoperiod)

Point 8: Line 271. Change ‘still existed” to persisted

Response 8: Thanks a lot, we have replaced “still existed” with “persisted”. (Line 276: persisted in the ovary)

Point 9: Line 276. after egg self-absorption – Change to egg resorption

Response 9: Thank you for pointing this out. we have replaced “egg self-absorption” with “egg resorption”.(Line 281: does not disappear after egg resorption)

Round 3

Reviewer 3 Report

I recommend that the authors find someone that can assist them in reading through and revising the manuscript.  Most of the changes I recommended in the two previous reviews were not carried out.  Please, pay attention to suggested revision below.

Manuscript Review – The effects of temperature on the development, morphology, and fecundity of Aenasius bambawalei (=Aenasius arizonensis

Remarks

Please, the manuscript is littered all over with the phrase responses and alteration.  This phrase does not make any meaning in most places that you inserted it.  

Specific comments

Summary and Abstract

I am NOT very comfortable with the phrase “responses and alteration”, which is found in several places in the summary, Abstract and Introduction.  The phrase does not convey the meaning of the changes that were observed.  If the changes were shortening of developing period, or enhanced development of the reproductive systems, or if high temperature induced egg resorption, these should be stated clearly.

Line 27-28: This is the first report of the impact of high temperatures on the pupal ----------------and fecundity of A. bambawalei.

Line 38. -------clarify induced developmental and morphological changes due to high ----------

Line 45. These findings will help understand adaptation mechanisms of --------------natural enemies.

Introduction

Line 57. And in this process, there may be some adaptative developmental and morphological changes occurring in the ----------

Line 61-62: ------ but not on the developmental and morphological changes resulting from elevated temperatures

Line 78: ----high temperature may cause some changes in development --------

Line 82: Replace responses and alteration with “adaptative”

Results

Discussion

The discussion should be revised in the light of this comment. The revision of the discussion demanded in the last review was not done.  Only a few changes were inserted at intervals.

Line 253: delete “more”

Author Response

Dear reviewer:

I'm awfully sorry that our revision failed to meet your requirements, and thank you for giving us the opportunity to revise again. According to your advise, we asked someone to help revise the article. For the comments put forward by the reviewers and not replied before, for example, the phrase “responses and alteration”, “global warming” and adaptive, we have made detailed replies in this revision in red or blue.

Point 1: Summary and Abstract 

I am NOT very comfortable with the phrase “responses and alteration”, which is found in several places in the summary and Abstract.  He phrase does not convey the meaning of the changes that were observed.  If the changes were shortening of developing period, or enhanced development of the reproductive systems, or if high temperature induced egg resorption, these should be stated clearly.  

Response 1: Special thanks to you for your good comments. Indeed, 

Indeed, the concept of "responses and alternation" is vague, which is not enough to explain the problem. Therefore, in the revision, we have specified different "responses" as follows: 

Line 13-14, Whether temperature changes can shorten or enhance the reproductive developmental period ---

Line 17-18, To understand the effect of temperature changes on the development of the reproductive systems of this parasitoid,

Line 31-32, ---whether this increase is the result of a shortened or enhanced development period of the reproductive systems of A. bambawalei ---

Line 34, --- to clarify the development and morphological changes induced by high te---

Line 52-53, ---In this process, some adaptive developmental, morphological and fecundity changes may occur in ---

Line 57-58, ---developmental, morphological or fecundity changes resulting from elevated temperatures ---

Line 59-60, ---changes in development, morphology and fecundity at high temperatures are exhibited ---

Line 73-74, ---high temperature may cause changes in the development, morphology and fecundity of A. bambawalei.

Point 2: Line 27-28: This is the first report of the impact of high temperatures on the pupal ----------------and fecundity of A. bambawalei.

Response 2: Thank you for pointing this out.  “This is the first report of the pupal development, morphology and fecundity alternation traits of A. bambawalei to high temperature .” has been replaced by “This is the first report of the impact of high temperature on the pupal development, morphology and fecundity alternation traits of A. bambawalei.” (Line23-24 , in blue). 

Point 3: Line 38. -------clarify induced developmental and morphological changes due to high ----------

Response 3: Considering the reviewer's suggestion, “--A. bambawalei under different temperatures to clarify its responses and alteration to high temperature and--” has been replaced by “ ---to clarify the development and morphological changes induced by high temperature and to better understand--- ”(Line34-35 , in blue) 

Point 4: Line 45. These findings will help understand adaptation mechanisms of --------------natural enemies.

Response 4: We have made correction according to the reviewer's comments, “These findings will help to clarify the responses and alteration mechanisms of accompanying natural enemies.” has been replaced by “These findings will help understand the adaptation mechanisms of accompanying natural enemies.” (Line41 , in blue) 

Point 5: Introduction

Line 57. And in this process, there may be some adaptative developmental and morphological changes occurring in the ----------

Response 5: Thank you for pointing this out. “And in this process, there may be some responses and alteration occurred in the --”

“In this process, some adaptive developmental, morphological and fecundity changes may occur in ---” (Line52-53 , in blue) 

Point 6: Line 61-62: ------ but not on the developmental and morphological changes resulting from elevated temperatures

Response 6: Thanks a lot, “but not on their responses and alteration to temperature --” has been corrected to “but not on the developmental, morphological or fecundity changes resulting from elevated temperatures ---”(Line56-58 , in blue) 

Point 7: Line 78: ----high temperature may cause some changes in development --------

Response 7: Thank you for pointing this out. “---that high temperature may cause some responses and alteration in development, morphology and fecundity of A. bambawalei.” has been corrected to “---high temperature may cause changes in the development, morphology and fecundity of A. bambawalei” (Line73-74, in blue) 

Point 8: Line 82: Replace “responses and alteration” with “adaptative”

Response 8: As the reviewer pointed out, it is more appropriate to replace “--- responses and alteration ---” with “---to clarifying the adaptive mechanism---” (Line77, in blue) 

Point 9: Discussion

It is generally known that high temperatures accelerate development in most exothermic animals including insects.  What is not fully understood is how global warming will impact or interface with general development of arthropods especially since global warming does not produce an even impact on all ecological habitats.  

Sometimes undue cold streams have been attributed to global warming.  Perhaps, adaptation to global warming can be assessed by comparing the responses of the same organisms over time, say 20 years.  A shift in those responses could be described as adaptation.

The discussion should be revised in the light of this comment. The revision of the discussion demanded in the last review was not done.  Only a few changes were inserted at intervals.

Response 9: Thank you for the patience and meticulous work. This part is the most controversial and the most revised part. For several core controversial points, we have revised the discussion. 

First, we have described in detail the development, morphology, reproduction and even specific structures such as egg handle hook analogues and compared with other species.

(Line 424-243, First, high temperature accelerates the growth and development rate of insects [3]. As expected, in our study, the temperature in the growth chamber ---

Line 255-256, Therefore, we also studied the effects of high temperature on other aspects of this parasitoid. 

Line 290-291, The more mature eggs there are at high temperature, the more oocyte stalk hook analogues are produced after egg resorption. )

Second, the global climate will not have a even impact on all organisms. It is inferred that there is the possibility of evolution by inferring the change of genotype of the host of A. bambawalei in the process of invasion and the close relationship between parasitoid and mealybug. In the process of invasion, the ambient temperature is rising. Based on the above reasons, we discussed that we will further study the genotype changes of A. bambawalei and its relationship with ambient temperature in the future.

(Line 268-280, ---- Phenotype is the result of the interaction of genetic factors and environmental factors (biological and abiotic). --- The genetic analysis of the North American and Asian groups of P. solenopsis shows that there has been great differentiation between the two groups. At the same time, the average temperature in China showed an increasing trend from 1961 to 2018, and the increasing trend rate in most areas exceeded 0.8°C/10a . Therefore, in the future, we need to verify whether the genotype of A. bambawalei has changed in different regions to speculate whether this parasitoid is adapting to global warming.)

Third, we pointed out our own shortcomings: we only studied the development, morphology and reproductive characteristics of short-term high temperature jumping wasp, which is not enough to fully explain its adaptability to global warming. However, this does not prevent us from speculating that the jumping wasp may have adaptability under the condition of global warming.

( Line 252-255, Of course, only the effects of high temperature on the pupal development duration and gonad development of A. bambawalei were studied, and the experiment was only  carried out for a short time, it is difficult to clarify the biological response of this parasitoid to rapid climate change. 

Line 322-325, Since global warming does not produce an even impact on all ecological habitats, how global warming will affect parasitoids and their role as natural enemies of pests is difficult to evaluate in the short term. However, the results of this study are still helpful for understanding the adaptive mechanisms of A. bambawalei.)

Point 10: Line 253: delete “more”

Response 10: Thanks for your comments, “---high temperatures is more conducive to A. bambawalei.” has been replaced by “--- high temperatures may be conducive to ---.” (Line 250-251)

    We tried our best to improve the manuscript and made some changes in the manuscript. These changes will not influence the content and framework of the paper. We appreciate for your warm work earnestly, and hope the correction will meet with approval.

    Once again, thank you very much for your comments and suggestions.